# Nutritional status and its associated factors among under five years Muslim children of Kapilvastu district, Nepal

**Chet Kant Bhusal** [1]*, **Sigma Bhattarai**[2], **Pradip Chhetri**[1], **Salau Din Myia**[3]

**1** Department of Community Medicine, Universal College of Medical Sciences and Teaching Hospital, Tribhuvan University, Bhairahawa, Rupandehi, Nepal, **2** Department of Nursing, Universal College of Medical Sciences and Teaching Hospital, Tribhuvan University, Bhairahawa, Rupandehi, Nepal, **3** Department of Public Health, CiST College, Sangamchock, Newbaneshor, Kathmandu, Nepal

* bhusalck3112@gmail.com

## Abstract

### Background

Malnutrition is a major public health problem throughout the world especially in Southeast Asia. This study aims to find out nutritional status and its associated factors among under five Muslim children of Kapilvastu district Nepal.

### Methods

Community based cross-sectional study was conducted among 336 under five Muslim children in Kapilvastu district Nepal from December 2021 to May 2022. Multistage probability random sampling was used. Among ten local units, three were selected randomly. Then from selected three units, two wards from each unit which covers large proportion of Muslim were selected purposively. After selecting wards, listing of household having children 6 to 59 months was done with the help of Female Community Health Volunteers and 56 children were selected by simple random sampling from each wards.

### Results

About half of Muslim children were underweight, 0.9% were overweight, 17.3% were wasted and 63.1% were stunted. Children with >4 members in family (AOR = 2.82, CI: 1.25–6.38), joint/extended family (AOR = 0.33, CI: 0.16–0.68), living with other than parents (AOR = 2.68, CI: 1.38–5.21), mother having primary (AOR = 2.59, CI: 1.09–6.10) and fathers having SLC and above education (AOR = 0.41, CI: 0.19–0.89), school going children (AOR = 0.27, CI: 0.15–0.48), no having agricultural land (AOR = 2.68, CI: 1.55–4.65), history of chronic diseases (AOR = 3.01, CI = 1.06–8.54) were significantly associated with underweight. Mothers having secondary (AOR = 0.30, CI: 0.10–0.88) and fathers having primary education (AOR = 3.50, CI: 1.26–9.74), school going children (AOR = 0.16, CI: 0.06–0.41), no having own land (AOR = 4.73, CI: 2.13–10.48), history of child chronic disease (AOR = 3.55, CI: 1.38–9.12) were significantly associated with wasting. Similarly, male children (AOR = 1.70, CI: 1.01–2.85), living in rural area (AOR = 0.17, CI: 0.09–0.31), joint/

**Data Availability Statement:** All relevant data are within the manuscript and its Supporting information files.

**Funding:** "NHRC Provincial Research Grant 2078/79" having Ref no. 1090 and protocol registration number 644/2021 P. The funders had no role in study design, data collection and analysis, decision to publish, or preparation of the manuscript.

**Competing interests:** The authors have declared that no competing interests exist.

**Abbreviations:** AOR, Adjusted odds ratio; CBS, Central Bureau of Statistics; CI, Confidence Interval; COPD, Chronic Obstructive Pulmonary Diseases; FCHV, Female Community Health Volunteer; HH, Household; M, Municipality; NDHS, Nepal Demographic and Health Survey; NHRC, Nepal Health Research Council; OR, Odds ratio; RM, Rural Municipality; SD, Standard Deviation; SDG, Sustainable Development Goal; SLC, School leaving certificate; SPSS, Statistical package for the social sciences; WHO, World Health Organization.

extended family (AOR = 0.28, CI: 0.13–0.64), living with other than parents (AOR = 3.71, CI: 1.84–7.49), fathers having secondary education (AOR = 0.50, CI: 0.27–0.94) and no having own land (AOR = 1.95, CI: 1.13–3.37) were significantly associated with stunting.

## Conclusions

Underweight, wasting and stunting in under-five Muslim children were above the cutoff point from the significant level of public health and higher than national data. Hence, this study suggests collaborative and immediate attention from responsible governmental and non-governmental organizations working in nutrition for providing informal learning opportunity, intervention regarding parental support to child, school enrolment at appropriate age, prevention and treatment of children's chronic diseases, intervention for income generating activities and addressing problems of household food insecurity among Muslim communities.

## 1. Introduction

Good nutrition is a prerequisite for the optimal health and growth, cognitive development, academic performance, productivity, socioeconomic development of individuals and national development of countries, however problems related to poor nutrition affect the entire population [1]. Malnutrition is a serious public health problem throughout the developing world predominantly in South Asian countries [2]. It is one of the most widespread causes of morbidity and mortality among children and adolescents throughout the world [3, 4]. Malnutrition in children is mainly due to unbalanced diet, inadequate or excessive food intake, non-potable water, poor and unhygienic food quality as well as sanitary practice [5–7], severe and repeated infectious diseases [7]. Malnutrition is a major contributing factor to disease, disability and early deaths for women and children [8–10]. Poverty, literacy and social status of mother are key factors contributing to malnutrition in children and this factors significantly contribute to the cause of propensity for malnutrition in an individual [11, 12]. Malnourished children who survive may suffer from recurrent illness and faltering growth, with permanent damage to their development and cognitive abilities [13]. Infants and young children bear the brunt of chronic malnutrition and suffer the greatest consequences, that is, the highest risks of morbidity and mortality [14]. Malnutrition early in life clearly has major consequences for future educational, income and productivity outcomes [15]. Adults who were malnourished during their early childhood have impaired intellectual performance [16]. Under nutrition during childhood followed by later overweight increases the risk of having non communicable diseases and for women increase the risk of childbirth complications [17]. Indicators of over nutrition such as overweight and obesity in children and adolescents now occur simultaneously with underweight, stunting and wasting [18–20]. The inconsistency of these two boundaries, repeatedly referred to as the "double burden of malnutrition" [20, 21].

Globally, in 2020, it was anticipated that 149 million children under the age of five were stunted, 45 million were wasted and 38.9 million were overweight or obese [22]. More than half of all stunted and more than two thirds of all wasted under-five children remained in Asia [23]. In South Asia, the prevalence of child wasting remains a serious public health issue as it exceeds the critical level of 15%. [24, 25]. The pooled prevalence of underweight and overweight in South Asia 28% and 17% respectively whereas in South East Asia both underweight

and overweight was 20% [26]. According to national level survey in Nepal, 36% of children under age 5 are stunted, 10% are wasted, 27% are underweight, and 1% are overweight [27]. The study conducted among under five years children in Dolakha and Kavre districts of Nepal revealed that 18.9%, 39.9% and 7.0% children were underweight, stunted and wasted respectively [28].

The Muslim population constitute of 4.4% of the overall population in Nepal, putting them in eighth position numerically. Rautahat, Bara, Parsa, Kapilvastu and Banke are the five districts which have highest number of Muslim populations [29]. The Muslim populations in Nepal was given a lower marginal status in the legal code, and they were shunned, discriminated against, and attacked violently [30]. Muslims, as a vulnerable demographic group, demanded social justice and the establishment of a Muslim branch of the National Inclusion Commission [31]. Muslim populations in Nepal have poor health outcomes and highest mortality rate [32]. Kapilvastu district of Lumbini province is one among the five districts of Nepal which have highest number (18.2%) of Muslim populations [29]. Malnutrition is a silent emergency and one of the most common causes of morbidity and mortality among children throughout the world especially in developing countries like Nepal. If unanswered, this challenge will endanger poverty reduction measures taken by governments, civil society, and aids based organizations and threaten their long-term growth prospects. In order to eliminate or reduce the risk of malnutrition, it is crucial to first understand and identify the causes and underlying factors of malnutrition. Among the Muslims community in Nepal in relation to health very few studies had been conducted which includes studies in family planning and maternal fertility [33–35]. However, according to researcher knowledge no research has been conducted to investigate the causes/reason of having malnutrition among Muslim children in Nepal. Hence this study aim to find out nutritional status and its associated factors among under 5 Muslim children of Kapilvastu district Nepal.

## 2. Materials and methods

### 2.1. Study design and source of population

Community based cross-sectional study was conducted among under 5 (6 months to 59 months) Muslim children in Kapilvastu district of Nepal from December 2021 to May 2022. Mothers of the Muslim children having age 6 to 59 months residing in the sampling area for more than one year was included for taking information regarding their children nutritional status, whereas their children were anthropometrically measured. In case of more than one children in same household, only one child was randomly selected using a lottery method and remaining children were excluded. Similarly, mothers of same age group children who had previous mental problem were excluded from the study.

### 2.2. Study setting

Kapilvastu district of Nepal is one amongst twelve district of Lumbini province of Nepal. Taulihawa one of the oldest city of Nepal is the Headquarter of this district. The district consists ten local units of which six are municipalities and four are rural municipalities. According to CBS 2011 district have total population of 571,936 of which 286,337 (50.10%) were females and 285,599 (49.90%) were males [29]. Kapilvastu is bounded by Rupandehi to the east, Dang district in the northwest, Arghakhanchi district to the north, Balrampur district, Awadh region, Uttar Pradesh, India to the west and Siddharthnagar district Purvanchal region, Uttar Pradesh to the south.

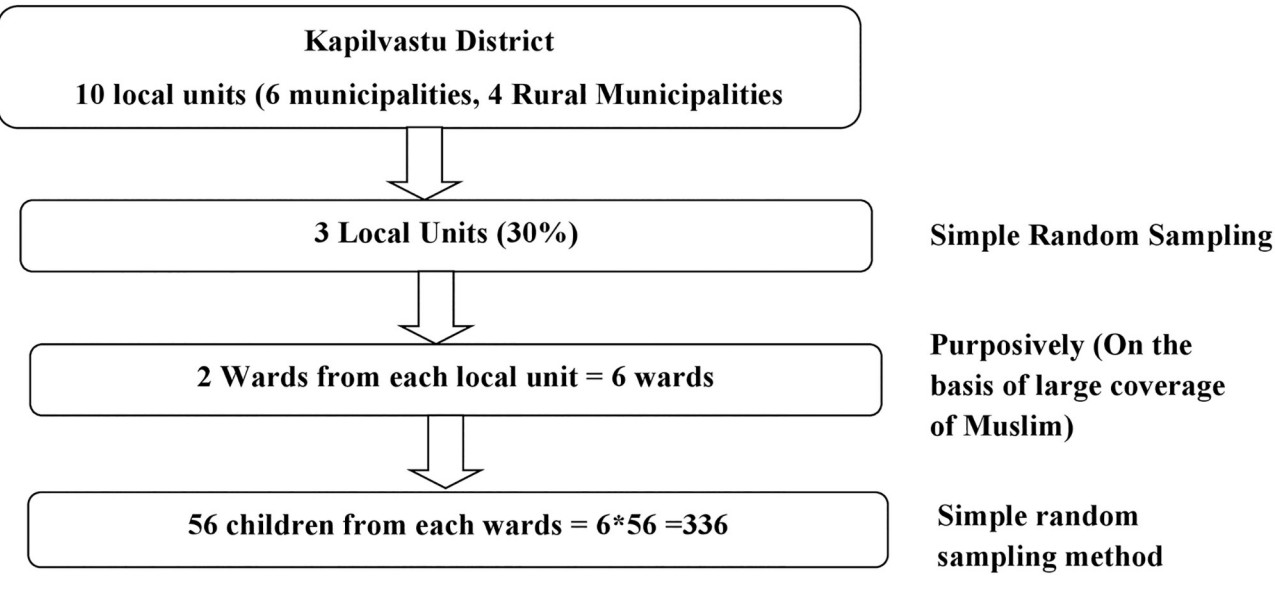

**Fig 1. Sampling technique.**

## 2.3. Sample size determination and sampling technique

The sample size was calculated by using the formula: $n = Z^2pq/L^2$ with 95% level of confidence interval, 6% margin of error and the prevalence of underweight was taken as 72.3% among the children in marginalized Chepang community of Chitwan district, Nepal [36]. As multistage probability random sampling technique was used, the calculated sample size 213 was multiplied by the design effect of 1.5. Hence the sample size was 213*1.5 = 319.5 that was 320. Additional 5% non-response rate was added in the above sample size; 5% of 320 = 16. Hence, final sample size of the study was 320 + 16 = 336. Multistage probability random sampling was used for the study. There are altogether ten local units (municipalities and rural municipalities) in Kapilvastu district of Nepal, out of which six are municipalities and four are rural municipalities. Among ten units, three units were selected randomly to generalize the Muslim community of the district. Then from selected three units, two wards from each local unit which covers a large proportion of Muslim population were selected purposively. After entering the selected wards, listing of the household (HH) having children of 6 months to 59 months in Muslim community was done from each ward with the help of Female Community Health Volunteers (FCHV)/community leaders. Than in order to select the respondents, equally fifty six mothers and their children of 6 months to 59 months from each selected wards were taken by using simple random sampling technique (Fig 1). Data were collected from mothers through face to face interview and anthropometric measurement of their children was done through door to door visit.

## 2.4. Data collection procedures and validity

A set of semi-structured questionnaire was formulated as data collection tool by reviewing different literatures [37–40] to collect the information of the mothers regarding their children nutritional status, food security status, environmental condition, feeding habits and so on. The questionnaire was translated into Nepali and Urdu then retranslated into English to find misinterpretation and correction were made. Pretesting of the questionnaire was done by taking

10% of total sample in wards number 3 of Siddharthnagar municipality of Rupandehi district, Nepal among Muslim children to ensure reliability of the study tools regarding its contents and information. In order to maintain the reliability of questionnaire, researcher used reliability analysis test through Cronbach's Alpha using pre-tested questionnaire. Fieldwork was carried out by enumerators on close monitoring and supervision by Principle Investigator and Co-Investigator. Five enumerators having qualification of Bachelor in Public Health and one Auxiliary Nursing Midwife with experience in research were trained for six days on data collection technique, probing technique, skipping pattern and ethical consideration. In each municipality, one Muslim enumerator was sent for data collection and translation. The collected data was reviewed and checked by principal investigator on regular basis.

## 2.5. Anthropometric measurement

The anthropometric measurement was done following the WHO guideline [41] in order to assess the nutritional status of the children. Seca digital scale was used for measuring weight and Stature meter was used for measuring height of children who were older than 24 months and can stand by themselves developmentally. Similarly, a hanging salter scale was used for measuring the weight and an inelastic measuring tape was used for measuring the height/length of the children who were less than equal to 24 months and were unable to stand by themselves. Less than equal to 24 months old children were measured while lying down. The weighing scale was standardized daily with standard weights. The weight was taken on barefoot and with minimal clothes to the nearest 0.1 kg while height was taken without shoes and cap to the nearest 0.1 cm. Age of the child was recorded by checking the birth certificate or vaccination card which was calculated by subtracting date of birth from date of interview.

## 2.6. Data processing and analysis

Collected data was followed by a cross check to ensure that all the questions have been filled correctly. Data checking, compiling and editing was done manually by the enumerators who were involved in the fieldwork. Collected data was entered into Microsoft excel, after that IBM SPSS version 22 was used to analyze the data. Initially descriptive analysis was performed as per the study variables. Simple frequency tables, cross tables and mean tables had been used to analyze data related to the study. WHO Anthro V 3.2.2 software [42] was used to calculate the anthropometric indices Z scores and the WHO 2006 cutoff points [43] was implemented to compare with the calculated Z-score and was classified in accordance to it. Children were considered as wasted if their weight for height/length Z-score (WHZ) is less than the reference population median by minus two standard deviations (-2SD). Children were considered as overweight if their weight for height/length Z-score (WHZ) is more than two standard deviations (+2SD) over the reference population's median. Similarly, children are considered as underweight if their weight for age Z-score (WAZ) is less than minus two standard deviations (-2SD) from the reference population's median. Bivariate analysis was carried out to find the association between independent variables and dependent variables by using chi square test. Odds ratio and 95% confidence interval was calculated to find out the significance of the association between study variables and outcome variables. Those variables which were significantly associated with dependent variable having p value < 0.05 in bivariate analysis was further analyzed using logistic regression model in multivariate analysis. Multicolinearity of the variables was checked before entering into the multivariate analysis. Adjusted odds ratio was calculated to measure the net effect of independent variables. Hosmer and Lemeshow test were used to test the goodness-of-fit for the regression models. The data were analyzed as per the objectives of the study.

## 2.7. Ethical approval and consent to participate

This study received ethical approval from Ethical Review Board of Nepal Health Research Council (Ref no. 1090 and protocol registration no.644/2021 P). Concerned stakeholders of municipalities and ward offices were officially contacted with letters and permission was obtained. After sharing the key information about the study written informed consent was taken from the participants whereas thumb print was taken from illiterate mothers and assent of children along with parent's informed consent was taken for mothers under the age of 18 years. Only those respondents who voluntarily agree to participate were involved in the study. The malnourished children were referred to nearby health center. All study participants were inform of their right to refuse participation and to leave the interview at any time. In order to reduce the stigmatization of religious issue, researcher followed different measures such as: one of the co- investigator is Muslim. Similarly in each municipality, one Muslim enumerator was sent for data collection and translation who had coordinated and collaborated with community and research team along with data collection. While approaching respondents, researcher called "Assalammualaikum" instead of Namaskar.

## 3. Results

The median age of the children was 39 months. Among 336 children, nearly one-fifth (19.3%) of the children were less than equal to 24 months. More than half (50.3%) of the mothers had more than two children. One-third of the respondents (33.3%) were from rural municipality. The median number of living children was three and size of family members was nine. Additionally, less than one fifth (19.3%) were from nuclear family. More than three fourth of the children were living with both mother and father in the family (Table 1).

Nearly one third (32.4%) of the child's mother and more than one fifth (20.8%) of child's father were illiterate. On the other hand, less than one fifth of both child's mother and father got informal education. More than one third (34.5) of the children started going to school. Regarding the occupation, 86.0% of childs' mother were involved in household chores and more than one fifth (21.4%) of their father were involved in agriculture. Less than half (48.2%) of the respondents had their own land and among them 41.3% had food sufficiency for only up to nine months (Table 2).

Table 3 represents the nutritional status of the under-five Muslim children in Kapilvastu district of Nepal. About half (50.3%) of the Muslim children were underweight, very few (0.9%) were overweight, 17.3% were wasted and nearly two third (63.1%) were stunted. The mean Z- score for weight for age (underweight) was -3.23 (95 percent CI: -3.369, -3.084), for weight for age (overweight) was 2.22 (95 percent CI: 1.940, 2.494), for weight for height (wasting) was -0.28 (95 percent CI: -0.505, -0.052) and for height for age (stunting) was -3.04 (95% CI:-3.254, -2.820) (Table 3).

Several factors were found significantly associated with underweight in the bivariate analysis as shown in Table 4. The multivariate regression analysis model identified size of family, type of family, living status of child, mother's education, father's education, child school going status, own agricultural land and history of child chronic disease as associated factors with underweight. Children who were living in family having more than four members were nearly 3 times more likely (AOR = 2.82, CI: 1.25–6.38) to have underweight than their counterpart. Similarly, odds of having underweight among children living in joint or extended family were 33% less likely (AOR = 0.33, CI: 0.16–0.68) than those living in nuclear family. Children who were living with other than both parents such as with only mother or only father or with their relatives were 2.68 times more likely (AOR = 2.68, CI: 1.38–5.21) to have underweight than those who were living with both parents. Children whose mother had primary level education;

**Table 1. Socio-demographic characteristics of mothers of under-five Muslim children in Kapilvastu district Nepal.**

| General Characteristics | Frequency (n = 336) | Percentage |
|---|---|---|
| **Age of Children** | | |
| ≤ 24 months | 65 | 19.3 |
| > 24 months | 271 | 80.7 |
| *Median age of children in months (Q₁ _Q₃); 39 (27–53)* | | |
| **Number of Living Children** | | |
| ≤ 2 children | 167 | 49.7 |
| > 2 children | 169 | 50.3 |
| *Median number of living children (Q₁ _Q₃); 3 (2–4)* | | |
| **Residential Status** | | |
| Urban (Municipality) | 224 | 66.7 |
| Rural (Rural Municipality) | 112 | 33.3 |
| **Size of Family** | | |
| ≤ 4 members | 46 | 13.7 |
| > 4 members | 290 | 86.3 |
| *Median Family Size (Q₁ _Q₃); 9 (6–12)* | | |
| **Type of Family** | | |
| Nuclear | 65 | 19.3 |
| Joint | 211 | 62.8 |
| Extended | 60 | 17.9 |
| **Living Status of Child** | | |
| Both parents | 259 | 77.1 |
| Only with Mother | 65 | 19.3 |
| Only with fathers | 1 | .3 |
| Other members in family | 11 | 3.3 |

that is who had grade one to five classes were 2.59 times more likely (AOR = 2.59, CI: 1.09–6.10) to have underweight. However, regarding father's education; those who had completed SLC and above education were 41% less likely (AOR = 0.41, CI: 0.19–0.89) to have underweight than those who were illiterate and had informal classes. School going children were 27% less likely (AOR = 0.27, CI: 0.15–0.48) to have underweight than non-school going children. The children of the respondents who did not have their own land were 2.68 times more likely (AOR = 2.68, CI: 1.55–4.65) to have underweight than those who had their own agricultural land. Children who had the history of chronic disease were 3.01 times more likely (AOR = 3.01, CI = 1.06–08.54) to have underweight than those who did not have history of chronic diseases. After subjected to multivariate model, father's occupation and history of parent's chronic disease were found as confounders for having underweight (Table 4).

After adjustment in multivariate analysis, variables such as mother's education, father's education, child school going status, having own agricultural land, family monthly income and children who had history of chronic disease are found as associated factors of wasting. Children whose mother had secondary level education (6 to 10 classes) were 30% less likely (AOR = 0.30, CI: 0.10–0.88) to be wasted than those who were illiterate and had informal classes. However, regarding father's education; those who had primary level education (1–5 classes) were 3.50 more likely (AOR = 3.50, CI: 1.26–9.74) to have wasting than those who were from illiterate and had informal classes. School going children were 16% less likely (AOR = 0.16, CI: 0.06–0.41) to have wasting than non-school going children. The children of the respondents who did not have their own land were 4.73 times more likely (AOR = 4.73, CI:

**Table 2. Socio-economic characteristics of mothers of under-five Muslim children in Kapilvastu district Nepal.**

| General Characteristics | Frequency (n = 336) | Percentage |
|---|---|---|
| **Mother's Education** | | |
| Illiterate | 109 | 32.4 |
| Informal Class | 55 | 16.4 |
| Primary (1–5 class) | 42 | 12.5 |
| Lower Secondary (6–8 class) | 41 | 12.2 |
| Secondary (9–10) | 44 | 13.1 |
| SLC | 32 | 9.5 |
| Higher Secondary or above | 13 | 3.9 |
| **Father's Education** | | |
| Illiterate | 70 | 20.8 |
| Informal Class | 56 | 16.7 |
| Primary (1–5 class) | 43 | 12.8 |
| Lower Secondary (6–8 class) | 47 | 14.0 |
| Secondary (9–10) | 56 | 16.7 |
| SLC | 33 | 9.8 |
| Higher Secondary and above | 31 | 9.2 |
| **Child School going status** | | |
| Not School going | 220 | 65.5 |
| School going | 116 | 34.5 |
| **Mother's Occupation** | | |
| Home maker | 289 | 86.0 |
| Business | 10 | 3.0 |
| Government or Private Job | 3 | .9 |
| Labour/Daily wages | 9 | 2.7 |
| Agriculture | 25 | 7.4 |
| **Father's Occupation** | | |
| Business | 78 | 23.2 |
| Government or Private Job | 35 | 10.4 |
| Labour/Daily Wages | 86 | 25.6 |
| Agriculture | 72 | 21.4 |
| Foreign Employee | 65 | 19.3 |
| **Own Agricultural Land** | | |
| Yes | 162 | 48.2 |
| No | 174 | 51.8 |
| **Food Sufficiency in a year from their own land (n = 162)** | | |
| Less than equal to 3 months | 14 | 8.6 |
| 4–6 months | 25 | 15.4 |
| 7–9 months | 28 | 17.3 |
| 10–12 months | 50 | 30.9 |
| More than 12 months (sell remaining food) | 45 | 27.8 |
| **Family Monthly Income (NRs)** | | |
| Less than 10000 | 120 | 35.7 |
| 10000–19999 | 101 | 30.1 |
| 20000–30000 | 56 | 16.7 |
| More than 30000 | 59 | 17.6 |

SLC- School Leaving Certificate

**Table 3. Prevalence of Nutritional Status of under-five Muslim children in Kapilvastu district Nepal.**

| Nutritional Status | Frequency (n = 336) | |
|---|---|---|
| | Yes n (%) | No n (%) |
| **Underweight** | 169 (50.3) | 167 (49.7) |
| Mean Z- score for weight for age (95% CI) | -3.23 (-3.369, -3.084) | |
| **Overweight** | 3 (0.9) | 333 (99.1) |
| Mean Z- score for weight for age (95% CI) | 2.22 (1.940–2.494) | |
| **Wasting** | 58 (17.3) | 278 (82.7) |
| Mean Z- score for weight for height (95% CI) | -0.28 (-0.505, -0.052) | |
| **Stunting** | 212 (63.1) | 124 (36.9) |
| Mean Z- score for height for age (95% CI) | -3.04 (-3.254, -2.820) | |

2.13–10.48) to have wasting than their counterparts. The children of the respondents with family monthly income more than NRs. 20,000 were 21% less likely (AOR = 0.21, CI: 0.08–0.57) to have wasting. Children who had history of chronic disease were 3.55 times more likely (AOR = 3.55, CI = 1.38–9.12) to be wasted. After subjected to multivariate model, age of children, number of living children, residential status and living status of children were found as confounders for having wasting (Table 5).

The bivariate analysis revealed several factors were significantly associated with stunting, as shown in Table 6. The multivariate logistic regression analysis model identified sex of child, residential status, type of family, living status of child, father's education and having own agricultural land as associated factors with stunting. Male were 1.7 times more likely (AOR = 1.70, CI: 1.01–2.85) to be stunted than female. The odds of having stunting were reduced by 17% (AOR = 0.17, CI = 0.09–0.31) among children who were living in rural areas than those who were living in urban areas. Regarding the type of family, the children who were living in joint and extended family were 28% (AOR = 0.28, CI = 0.13–0.64) less likely to be stunted than those who were living in nuclear family. Similarly, the odds of having stunting were 3.71 times more likely (AOR = 3.71, CI = 1.84–7.49) among the children who lived with other than both parents (either with only mothers or with only fathers or with relatives) than those who were living with both parents. The odds of having stunting were reduced by 50% (AOR = 0.50, CI = 0.27–0.94) among those children whose mothers had secondary education than those who were illiterate or had informal classes. Children who were from the family who did not have their own agricultural land were 1.95 times more likely (AOR = 1.95, CI = 1.13–3.37) to be stunted than those who had their own agricultural land (Table 6).

## 4. Discussion

The finding of this study revealed that 50.3% of the under-five Muslim children were underweight, 17.3% were wasted, 0.9% were overweight and 63.1% were stunted. The findings of this study were above the cutoff point from the significant level of public health i.e ≥ 30% for underweight and ≥ 40% for stunting and ≥ 15% for wasting [44, 45]. The under nutritional indicators such as underweight, wasting and stunting in this study was higher than in the national level data, however over nutritional indicators (overweight) is similar with Nepal Demographic and Health Survey 2016 [27]. These discrepancy might be due to the variances in dietary intake, socioeconomic and sociocultural factors prevalent in the marginalized Muslim children. The study conducted in Janakpurdham, Nepal among 161 Muslim people found that Muslim communities in Nepal have poor health outcomes and highest mortality rate [32].

**Table 4. Factors associated with underweight in using bivariate and multivariate regression model.**

| Characteristics | Underweight Status (%) | | p-value | COR (95% CI) | AOR (95% CI) |
|---|---|---|---|---|---|
| | No (n = 167) | Yes (n = 169) | | | |
| **Age of Children** | | | | | |
| ≤ 24 months | 27(16.2) | 38(22.5) | 0.143 | 1 | Ns |
| > 24 months | 140(83.8) | 131(77.5) | | 0.67(0.38–1.15) | |
| **Number of Living Children** | | | | | |
| ≤ 2 children | 74(44.3) | 93(55.0) | 0.050 | 1 | Ns |
| > 2 children | 93(55.7) | 76(45.0) | | 0.65(0.42–1.00) | |
| **Residential Status** | | | | | |
| Urban | 112(67.1) | 112(66.3) | 0.877 | 1 | Ns |
| Rural | 55(32.9) | 57(33.7) | | 1.04(0.66–1.63) | |
| **Size of Family** | | | | | |
| ≤ 4 members | 30(18.0) | 16(9.5) | 0.023* | 1 | 1 |
| > 4 members | 137(82.0) | 153(90.5) | | **2.09(1.09–4.01)** | **2.82(1.25–6.38)** |
| **Type of Family** | | | | | |
| Nuclear | 23(13.8) | 42(24.9) | 0.010* | 1 | **1** |
| Joint and Extended | 144(86.2) | 127(75.1) | | **0.48(0.28–0.85)** | **0.33(0.16–0.68)** |
| **Marital Status** | | | | | |
| Married | 156(93.4) | 157(92.9) | 0.852 | 1 | Ns |
| Divorced, Single or widow/widower | 11(6.6) | 12(7.1) | | 1.08(0.46–2.53) | |
| **Living Status of Child** | | | | | |
| Both parents | 145(86.8) | 114(67.5) | <0.001** | 1 | 1 |
| Other than both parents | 22(13.2) | 55(32.5) | | **3.18(1.83–5.52)** | **2.68(1.38–5.21)** |
| **Mother's Education** | | | | | |
| Illiterate and Informal class | 76(45.5) | 88(52.1) | 0.036* | 1 | 1 |
| Primary | 16(9.6) | 26(15.4) | | 1.40(0.70–2.81) | **2.59(1.09–6.10)** |
| Secondary | 45(26.9) | 40(23.7) | | 0.77(0.45–1.30) | 1.43(0.72–2.84) |
| SLC and above | 30(18.0) | 15(8.9.0) | | **0.43(0.22–0.86)** | 0.81(0.35–1.88) |
| **Father's Education** | | | | | |
| Illiterate and Informal class | 51(30.5) | 75(44.4) | 0.008* | 1 | 1 |
| Primary | 18(10.8) | 25(14.8) | | 0.94(0.47–1.91) | 0.83(0.36–1.91) |
| Secondary | 57(34.1) | 46(27.2) | | **0.55(0.32–0.93)** | 0.54(0.28–1.05) |
| SLC and above | 41(24.6) | 23(13.6) | | **0.38(0.21–0.71)** | **0.41(0.19–0.89)** |
| **Child School going status** | | | | | |
| Not School going | 93(55.7) | 127(75.1) | <0.001** | 1 | 1 |
| School going | 74(44.3) | 42(24.9) | | **0.42(0.26–0.66)** | **0.27(0.15–0.48)** |
| **Mother's Occupation** | | | | | |
| Home maker | 145(68.8) | 144(85.2) | 0.669 | 1 | Ns |
| Other than home maker | 22(13.2) | 25(14.8) | | 1.14(0.62–2.12) | |
| **Father's Occupation** | | | | | |
| Agriculture | 44(26.3) | 28(16.6) | 0.029* | 1 | 1 |
| Other than Agriculture | 123(73.7) | 141(83.4) | | **1.80(1.06–3.07)** | 1.38(0.74–2.59) |
| **Own Agricultural Land** | | | | | |
| Yes | 103(61.7) | 59(34.9) | <0.001** | 1 | 1 |
| No | 64(38.3) | 110(65.1) | | **3.00(1.92–4.68)** | **2.68(1.55–4.65)** |
| **History of child chronic disease** | | | | | |
| No | 162(97.0) | 154(91.1) | 0.023 | 1 | 1 |
| Yes | 5(3.0) | 15(8.9) | | **3.16(1.12–8.89)** | **3.01(1.06–8.54)** |

(*Continued*)

**Table 4.** (Continued)

| Characteristics | Underweight Status (%) | | p-value | COR (95% CI) | AOR (95% CI) |
|---|---|---|---|---|---|
| | No (n = 167) | Yes (n = 169) | | | |
| **History of parents' chronic disease** | | | | | |
| No | 142(85.0) | 128(75.7) | 0.032 | 1 | 1 |
| Yes | 25(15.0) | 41(24.3) | | **1.82(1.05–3.16)** | 1.77(1.01–3.08) |

*Significant at p<0.05,

**highly significant at p <0.001,

1 = Reference category, Ns = not significant in bivariate analysis

The finding of this study revealed that half of the children were underweight which is in line with another community based cross-sectional study conducted among two hundred ninety two under five children in Rupandehi district of Nepal which found that nearly half (45.9%) of the children were underweight [40]. However, in contrast to this study, several other studies conducted in Nepal such as study conducted by analyzing 12, 674 data from Nepal Demographic and Health Survey (NDHS 2011) focusing severely food-insecure households in Nepal [46], rural and urban Nepal [28, 47–52]. Similarly, several studies conducted in India such as community-based cross-sectional study conducted among 500 children of tribal population in Assam India [53], community-based cross-sectional study conducted at urban slums and rural area in two districts of Maharashtra, India among 3671 under five children [54]. Similar studies conducted around the world such as cross-sectional study conducted at semi-pastoral communities of Tanzania among 310 children of 6 to 59 months [7], community based cross sectional study conducted in eastern Ethiopia among 791 under five years children [55] found lower proportion of under five children were underweight. However, higher proportion of underweight among under five children was also found in some studies, such as study case-control conducted at a Teaching Hospital in Bobo-Dioulasso, Burkina Faso among 164 HIV infected and 164 HIV uninfected under five children found more than three fourth of HIV infected children were underweight [56]. The prevalence of overweight in this study is found to be lower than another community based descriptive cross-sectional study conducted in Chitwan district of Nepal among 167 under five children which found 5.39% were overweight [50]. The prevalence of wasting in the present study was nearly one fifth which is in line with another community-based cross-sectional study conducted at urban slums and rural area of Maharashtra, India [54] and secondary analysis of Nigeria Demographic and Health Survey NDHS, Nigeria [57]. In contrast to this study, several other studies conducted in Nepal [48–50], cross-sectional study conducted at semi-pastoral communities of Tanzania among 310 children [7], eastern Ethiopia among 791 [55] and community based cross sectional study conducted in Sudan among 411 under five children [4] found lower proportion of under five children were wasted. However, in contrast to this study, several other studies conducted in Nepal [47, 51], community based cross-sectional study conducted among 500 children of tribal population in Assam India [53], cross- sectional household survey conducted among 1635 under five children in North Sudan [58] and case-control study conducted taking 164 HIV infected and 164 HIV uninfected under five years children in Burkina Faso found more than one fifth of the children were wasted [56]. Nearly two third of the children in this study were stunted which is in line with another study conducted in Rupandehi [40] and Kapilvastu district of Nepal [47]. However, in contrast to this study other studies such as study conducted in severely food-insecure households in Nepal [46], rural and urban Nepal [28, 48–51], tribal population in Assam India [53], urban slums and rural area of Maharashtra, India [54], semi-pastoral

**Table 5. Factors associated with wasting in using bivariate and multivariate regression model.**

| Characteristics | Wasting Status (%) | | p-value | COR (95% CI) | AOR (95% CI) |
|---|---|---|---|---|---|
| | No (n = 278) | Yes (n = 58) | | | |
| **Age of Children** | | | | | |
| ≤ 24 months | 46(16.5) | 19(32.8) | 0.004* | 1 | 1 |
| > 24 months | 232(83.5) | 39(67.2) | | **0.41(0.22–0.77)** | 0.47(0.21–1.06) |
| **Number of Living Children** | | | | | |
| ≤ 2 children | 131(47.1) | 36(62.1) | 0.038* | 1 | 1 |
| > 2 children | 147(52.9) | 22(37.9) | | **0.55(0.31–0.97)** | 0.61(0.30–1.25) |
| **Residential Status** | | | | | |
| Urban | 194(69.8) | 30(51.7) | 0.008* | 1 | 1 |
| Rural | 84(30.2) | 28(48.3) | | **2.16(1.21–3.83)** | 1.24(0.59–2.60) |
| **Size of Family** | | | | | |
| ≤ 4 members | 41(14.7) | 5(8.6) | 0.217 | 1 | Ns |
| > 4 members | 237(85.3) | 53(91.4) | | 1.83(0.69–4.86) | |
| **Type of Family** | | | | | |
| Nuclear | 51(18.3) | 14(24.1) | 0.310 | 1 | Ns |
| Joint and Extended | 227(81.7) | 44(75.9) | | 0.71(0.36–1.39) | |
| **Living Status of Child** | | | | | |
| Both parents | 221(79.5) | 38(65.5) | 0.021* | 1 | 1 |
| Other than both parents | 57(20.5) | 20(34.5) | | **2.04(1.10–3.77)** | 1.52(0.67–3.48) |
| **Mother's Education** | | | | | |
| Illiterate and Informal class | 126(45.3) | 38(65.5) | 0.038 | 1 | 1 |
| Primary | 36(12.9) | 6(10.3) | | 0.55(0.22–1.41) | 0.59(0.19–1.82) |
| Secondary | 77(27.7) | 8(13.8) | | **0.34(0.15–0.78)** | **0.30(0.10–0.88)** |
| SLC and above | 39(14.0) | 6(10.3) | | 0.51(0.20–1.30) | 1.11(0.35–3.49) |
| **Father's Education** | | | | | |
| Illiterate and Informal class | 105(37.8) | 21(36.2) | 0.004* | 1 | 1 |
| Primary | 28(10.1) | 15(25.9) | | **2.68(1.22–5.86)** | **3.50(1.26–9.74)** |
| Secondary | 86(30.9) | 17(29.3) | | 0.99(0.49–1.99) | 2.03(0.80–5.15) |
| SLC and above | 59(21.2) | 5(8.6) | | 0.42(0.15–1.18) | 0.67(0.19–2.34) |
| **Child School going status** | | | | | |
| Not School going | 169(60.8) | 51(87.9) | <0.001* | 1 | 1 |
| School going | 109(39.2) | 7(12.1) | | **0.21(0.93–0.49)** | **0.16(0.06–0.41)** |
| **Mother's Occupation** | | | | | |
| Home maker | 239(86.0) | 50(86.2) | 0.962 | 1 | Ns |
| Other than home maker | 39(14.0) | 8(13.8) | | 0.98(0.43–2.23) | |
| **Father's Occupation** | | | | | |
| Agriculture | 63(22.7) | 9(15.5) | 0.291 | 1 | Ns |
| Other than Agriculture | 215(77.3) | 49(84.5) | | 1.60(0.74–3.43) | |
| **Own Agricultural Land** | | | | | |
| Yes | 150(54.0) | 12(20.7) | <0.001* | 1 | 1 |
| No | 128(46.0) | 46(79.3) | | **4.49(2.28–8.85)** | **4.73(2.13–10.48)** |
| **Family Monthly Income** | | | | | |
| <10000 (NRs) | 93(33.5) | 27(46.6) | 0.002* | 1 | 1 |
| 10000–20000 (NRs) | 78(28.1) | 23(39.7) | | 1.02(0.54–1.91) | 0.68(0.31–1.49) |
| >20000 (NRs) | 107(38.5) | 8(13.8) | | **0.26(0.11–0.59)** | **0.21(0.08–0.57)** |
| **History of child chronic disease** | | | | | |
| No | 266(95.7) | 50(86.2) | 0.006 | 1 | 1 |

(*Continued*)

**Table 5.** (Continued)

| Characteristics | Wasting Status (%) | | p-value | COR (95% CI) | AOR (95% CI) |
|---|---|---|---|---|---|
| | No (n = 278) | Yes (n = 58) | | | |
| Yes | 12(4.3) | 8(13.8) | | **3.55(1.38–9.12)** | **3.55(1.38–9.12)** |
| History of parents' chronic disease | | | | | |
| No | 225(80.9) | 45(77.6) | 0.559 | 1 | Ns |
| Yes | 53(19.1) | 13(22.4) | | 1.23(0.62–2.44) | |

*Significant at p<0.05,

**highly significant at p <0.001,

1 = Reference category, Ns = not significant in bivariate analysis

communities of Tanzania, eastern Ethiopia [55] found lower proportion of under five children were stunted [7]. However, higher proportion of underweight was also found in some studies, such as study conducted at a Teaching Hospital in Bobo-Dioulasso, Burkina Faso found more than three fourth of HIV infected under five years children were stunted [56].

In the present study, the prevalence of under nutrition particularly underweight was greater than that of wasting. Similar results were also found in other studies such as study conducted in Nepal [47, 59]. tribal population in Assam India [53], urban slums and rural area of Maharashtra, India [54]. Underweight was shown to be more prevalent than wasting in this study, which is in line with several other studies such as study Nepal [47, 50, 59], tribal population in Assam India [53], urban slums and rural area of Maharashtra, India [54], Eastern Ethiopia [55]. These divergence on the prevalence of under nutrition might be due to the variances in study population, study setting, and dietary intake as well as socioeconomic and cultural factors. This study was conducted in marginalized Muslim children, and Muslim populations in Nepal have poor health outcomes and highest mortality rate [32].

Size of family was significantly associated with underweight in this study, which is in line with another study conducted among tribal population in Assam India [53], cross-sectional descriptive study conducted in Kenya among 144 children [60] cross- sectional household survey conducted among 325 children in, Ethiopia [61] and north Sudan [58] which founds that underweight is more prevalent in the children living in greater than four members in the family. The present study revealed that type of family had statistically significant association with the underweight. This result is consistent with result of study conducted in Chitwan district of Nepal [50]. In this study children living with other than both the parents were more likely to be underweight than their counterparts, this results is similar to another study conducted North Ethiopia [61] which found that children not living with their both parents were 3.6 times more likely to be underweight. Similarly another study conducted by analyzing data from demographic and health surveys of 17 Sub-Saharan Africa found that overall malnutrition is more prevalent in west and Sub-Saharan African under five children who were not living with any of the parents [62]. In the present study, mother's education was significantly associated with underweight of under five children. This result is consistent with result of study conducted among under five children residing in rural area of Maharashtra, India [54], Nigeria [57], community based cross-sectional survey conducted in Iran [63], Indonesia [64] and south Ethiopia [65]. Children's from the fathers having SLC and above education were less likely to be underweight than from illiterate fathers which is in accordance to the finding of other studies such as study from Nigeria [57] and Sri Lanka [66] which found that fathers having secondary and above education were less likely to be underweight than illiterate fathers. In the present study ownership of own agricultural land is positively associated with having

**Table 6. Factors associated with Stunting in using bivariate and multivariate regression model.**

| Characteristics | Stunting Status (%) | | p-value | COR (95% CI) | AOR (95% CI) |
|---|---|---|---|---|---|
| | No (n = 124) | Yes (n = 212) | | | |
| **Age of Children** | | | | | |
| ≤ 24 months | 27(21.8) | 38(17.9) | 0.389 | 1 | Ns |
| > 24 months | 97(78.2) | 174(82.1) | | 1.28(0.73–2.21) | |
| **Sex of Child** | | | | | |
| Female | 71(57.3) | 96(45.3) | 0.034* | 1 | 1 |
| Male | 53(42.7) | 116(54.7) | | **1.62(1.04–2.53)** | **1.70(1.01–2.85)** |
| **Number of Living Children** | | | | | |
| ≤ 2 children | 61(49.2) | 106(50.0) | 0.910 | 1 | Ns |
| > 2 children | 63(50.8) | 106(50.0) | | 0.97(0.62–1.51) | |
| **Residential Status** | | | | | |
| Urban | 63(50.8) | 161(75.9) | <0.001* | 1 | 1 |
| Rural | 61(49.2) | 51(24.1) | | **0.33(0.20–0.53)** | **0.17(0.09–0.31)** |
| **Size of Family** | | | | | |
| ≤ 4 members | 24(19.4) | 22(10.4) | 0.021* | 1 | 1 |
| > 4 members | 100(80.6) | 190(89.6) | | **2.07(1.11–3.88)** | 2.3(0.99–5.36) |
| **Type of Family** | | | | | |
| Nuclear | 15(12.1) | 50(23.6) | 0.010* | 1 | 1 |
| Joint and Extended | 109(87.9) | 162(76.4) | | **0.45(0.24–0.83)** | **0.28(0.13–0.64)** |
| **Living Status of Child** | | | | | |
| Both parents | 108(87.1) | 151(71.2) | 0.001 | 1 | 1 |
| Other than both parents | 16(12.9) | 61(28.8) | | **2.73(1.49–4.99)** | **3.71(1.84–7.49)** |
| **Mother's Education** | | | | | |
| Illiterate and Informal class | 58(46.8) | 106(50.0) | 0.782 | 1 | Ns |
| Primary | 14(11.3) | 28(13.2) | | 1.09(0.53–2.24) | |
| Secondary | 35(28.2) | 50(23.6) | | 0.78(0.46–1.34) | |
| SLC and above | 17(13.7) | 28(13.2) | | 0.90(0.46–1.78) | |
| **Father's Education** | | | | | |
| Illiterate and Informal class | 39(31.5) | 87(41.0) | 0.033 | 1 | 1 |
| Primary | 14(11.3) | 29(13.7) | | 0.93(0.44–1.95) | 1.06(0.47–2.41) |
| Secondary | 50(40.3) | 53(25.0) | | **0.48(0.28–0.82)** | **0.50(0.27–0.94)** |
| SLC and above | 21(16.9) | 43(20.3) | | 0.92(0.48–1.75) | 0.67(0.32–1.41) |
| **Child School going status** | | | | | |
| Not School going | 79(63.7) | 141(66.5) | 0.602 | 1 | Ns |
| School going | 45(36.3) | 71(33.5) | | 0.88(0.56–1.41) | |
| **Mother's Occupation** | | | | | |
| Home maker | 105(84.7) | 184(86.8) | 0.590 | 1 | Ns |
| Other than home maker | 19(15.3) | 28(13.2) | | 0.84(0.45–1.58) | |
| **Father's Occupation** | | | | | |
| Agriculture | 26(21.0) | 46(21.7) | 0.875 | 1 | Ns |
| Other than Agriculture | 98(79.0) | 166(78.3) | | 0.96(0.56–1.65) | |
| **Own Agricultural Land** | | | | | |
| Yes | 69(55.6) | 93(43.9) | 0.037* | 1 | 1 |
| No | 55(44.4) | 119(56.1) | | **1.61(1.03–2.51)** | **1.95(1.13–3.37)** |
| **History of child chronic disease** | | | | | |
| No | 119(96.0) | 197(92.9) | 0.255 | 1 | Ns |
| Yes | 5(4.0) | 15(7.1) | | 1.81(0.64–5.11) | |

*(Continued)*

**Table 6.** (Continued)

| Characteristics | Stunting Status (%) | | p-value | COR (95% CI) | AOR (95% CI) |
|---|---|---|---|---|---|
| | No (n = 124) | Yes (n = 212) | | | |
| **History of parents' chronic disease** | | | | | |
| No | 101(81.5) | 169(79.7) | 0.699 | 1 | Ns |
| Yes | 23(18.5) | 43(20.3) | | 1.12(0.64–1.96) | |

*Significant at p<0.05,

**highly significant at p <0.001,

1 = Reference category, ns = not significant in bivariate analysis

children underweight and wasting which is similar to the review conducted for finding the nutritional status of south Asia [67]. The present study revealed that child suffering from chronic diseases were more likely to be underweight which is in line with another study conducted by pooled analysis of 2416 population-based measurement studies [68]. Children's from the mother having secondary education were less likely to be wasted than from illiterate mothers which is in accordance to the finding of another study from Nigeria [57] which found that mothers having secondary and above education were 0.79 times less likely to be wasted than illiterate mothers. In this study father's education is also found to be significantly associated with wasting, this finding is in accordance to the finding of another study from Nigeria [57]. Family monthly income in this study was found to be significantly associated with wasting which is in concordance with another cross-sectional descriptive study conducted among 99 under five children in Zambia [69].

The present study showed that male were 1.70 times more likely to be stunted than female, similar result was also found in other studies such as study conducted in urban slums area of Maharastra India [54], slums area of Nairobi [70], cross-sectional study conducted among 341 under five children in Tanzania [71] and community based cross-sectional study conducted in south Ethiopia among 796 children [65] where male were 1.77, 1.60, 1.89 and 2.8 times more likely to be stunted than their counterparts respectively. Similarly female were 0.82 times less likely to be stunted in another study conducted in Ethiopia by using 2016 Ethiopia demographic and health survey [72]. Residential status in this study was significantly associated with stunting which is in concordance with several other studies [51, 63, 64, 71, 73]. In the present study stunting was less prevalent in the children who were living in joint and extended family, however in contrast to this study another study conducted in urban slums of Maharastra India [54] found less prevalent of stunting among children living in nuclear family. Similarly, stunting in under five children is more prevalent among those mothers who do not have their own agriculture land, this result is similar to the result from Zambia [74]. In the present study having own agriculture land is significantly associated with stunting which is line with another descriptive cross-sectional study conducted among 423 under five children in Nigeria, where food insecure mothers were more likely to be stunted [75]. This variations might be due to variations in study populations, dietary diversity, socio cultural and socio economic factors.

## 5. Conclusions

The key nutritional indicators such as underweight, wasting and stunting in under-five Muslim children were above the cutoff point from the significant level of public health given by WHO, as well as higher than the national level data from Nepal Demographic and Health Survey 2016 hence this study suggests a collaborative and immediate attention should be taken

from the responsible governmental and non-governmental organizations such as local government, municipalities, rural municipality, local NGOs/INGOs and public health authorities working in nutrition and health.

Children having greater than four members in the family, nuclear family, who are living with other than their parents, mother's having primary education, illiterate fathers, children who were not going school, who do not have their own agricultural land and history of child's chronic disease were found to be independent predictors of increased risks for underweight. Illiterate mothers, father's having primary education, children who were not going school, who do not have their own agricultural land, family monthly income of lesser than 10000 Nepalese rupees and history of child's chronic disease were found to be independent predictors of increased risks for wasting. Similarly, male children, living in urban area, nuclear family, who are living with other than their parents, having illiterate fathers, who do not have their own agricultural land were found to be independent predictors of increased risks for stunting. Based on these findings, efforts should be done to promote the women and men education by providing informal learning opportunity, intervention regarding parental support to child, children school enrolment at appropriate age, prevention and treatment of children chronic diseases, intervention for income generating activities and intervention addressing the problems of household food insecurity should be emphasized among the Muslim communities. Similarly, large-scale research within Muslim communities should be recommended using qualitative approaches to investigate perceptions cultural practice, belief, and value system related to nutritional status among under five children of underprivileged and unserved Muslim communities.

## 6. Strengths and limitations of the study

This study may have a few limitations that might affect the validity of the results. The cross-sectional study design restricts the ability to examine the temporal relationship of the findings. Besides, information bias is possible as the respondents had to recall some of the information regarding independent variables. This study has a significant asset in that it was conducted in a specific marginalized Muslim community, where information about nutritional status is very scare. Besides, it was community based study in which interview and anthropometric measurement was done by house to house visit, it represent the real Muslim community.

### 6.1. Implications for national program and health services

Sustainable Development Goal 3 (SDG 3) target to reduce neonatal mortality to at least as low as 12 per 1000 live births and under-5 mortality to at least as low as 25 per 1000 live births. Among the total childhood mortality 53% is due to malnutrition. Although the Nepalese government had attempted to incorporate Socially Inclusive and Marginalized Groups in various policies and plans, there is a scarcity of data/information among marginalized populations such as Muslims. The outcomes of this study demonstrated the importance of developing strategic behavior communication programs for disadvantaged groups, such as Muslims.

### 6.2. Areas of further research

Further studies need to be done in a large scale to explore cultural practice, belief, and value system regarding nutritional status of under five children among disadvantaged and marginalized communities such as Muslim.

## Supporting information

**S1 File.**
(SAV)

## Acknowledgments

Authors would like to express their deepest gratitude to public health graduates Mr. Sudarshan Gautam, Adolescent and youth Program Officer in Kapilvastu Integrated Development Services, Mr. Rajan Gupta NAMUNA Integrated Development Council (NAMUNA) District Coordinator/Field Supervisor for support in field level coordination and data collection, Mr. Jaan Mohammad Pathan, BPH 4th year student, Ms. Madhu Chaudary and Ms. Kumari Sweta Gupta Kalwar BPH 3rd year students and Ms. Pooja Gupta, Adolescents Youth Community Mobilizer in Kapilvastu Integrated Development Services for their support in data collection. Last but not least, we would like to thank all the respondent mothers and children for their valuable time and support for providing inevitable information for the study.

## Author Contributions

**Conceptualization:** Chet Kant Bhusal, Sigma Bhattarai.

**Data curation:** Chet Kant Bhusal.

**Formal analysis:** Chet Kant Bhusal, Pradip Chhetri.

**Funding acquisition:** Chet Kant Bhusal.

**Investigation:** Chet Kant Bhusal, Sigma Bhattarai, Pradip Chhetri, Salau Din Myia.

**Methodology:** Chet Kant Bhusal.

**Project administration:** Chet Kant Bhusal, Salau Din Myia.

**Resources:** Chet Kant Bhusal, Salau Din Myia.

**Software:** Chet Kant Bhusal.

**Supervision:** Chet Kant Bhusal.

**Validation:** Chet Kant Bhusal, Sigma Bhattarai.

**Visualization:** Chet Kant Bhusal.

**Writing – original draft:** Chet Kant Bhusal.

**Writing – review & editing:** Chet Kant Bhusal, Sigma Bhattarai, Pradip Chhetri, Salau Din Myia.

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
