## [Decision Letter · Decision Letter 0]

29 Nov 2022

PONE-D-22-29132Nutritional Status and Its Associated Factors Among Under Five Years Muslim Children of Kapilvastu District, NepalPLOS ONE

Dear Dr. BHUSAL,

Thank you for submitting your manuscript to PLOS ONE. After careful consideration, we feel that it has merit but does not fully meet PLOS ONE’s publication criteria as it currently stands. Therefore, we invite you to submit a revised (minor)  version of the manuscript that addresses the points raised during the review process. One of the major concerns of both the reviewers was regarding reporting of study findings in standard English. Thus, amend the manuscript accordingly.  Please submit your revised manuscript by Jan 13 2023 11:59PM. If you will need more time than this to complete your revisions, please reply to this message or contact the journal office at plosone@plos.org. Please include the following items when submitting your revised manuscript:A rebuttal letter that responds to each point raised by the academic editor and reviewer(s). You should upload this letter as a separate file labeled 'Response to Reviewers'.A marked-up copy of your manuscript that highlights changes made to the original version. You should upload this as a separate file labeled 'Revised Manuscript with Track Changes'.An unmarked version of your revised paper without tracked changes. You should upload this as a separate file labeled 'Manuscript'.If applicable, we recommend that you deposit your laboratory protocols in protocols.io to enhance the reproducibility of your results. Protocols.io assigns your protocol its own identifier (DOI) so that it can be cited independently in the future. For instructions see: https://journals.plos.org/plosone/s/submission-guidelines#loc-laboratory-protocols. Additionally, PLOS ONE offers an option for publishing peer-reviewed Lab Protocol articles, which describe protocols hosted on protocols.io. Read more information on sharing protocols at https://plos.org/protocols?utm_medium=editorial-email&utm_source=authorletters&utm_campaign=protocols.

We look forward to receiving your revised manuscript.

Kind regards,

Reshu Agrawal Sagtani

Academic Editor

PLOS ONE

Journal Requirements:

"“NHRC Provincial Research Grant 2078/79” having Ref no. 1090 and protocol registration no.644/2021 P"

"Authors would like to express their deepest gratitude to Nepal Health Research Council (NHRC Nepal) for providing “NHRC Provincial Research Grant 2078/79” having protocol number (64412021 P) for conducting this study.."

"“NHRC Provincial Research Grant 2078/79” having Ref no. 1090 and protocol registration no.644/2021 P"

Reviewers' comments:

Reviewer's Responses to Questions

**Comments to the Author**

1. Is the manuscript technically sound, and do the data support the conclusions?

Reviewer #1: Yes

Reviewer #2: Yes

2. Has the statistical analysis been performed appropriately and rigorously? 

Reviewer #1: Yes

Reviewer #2: Yes

3. Have the authors made all data underlying the findings in their manuscript fully available?

Reviewer #1: Yes

Reviewer #2: Yes

4. Is the manuscript presented in an intelligible fashion and written in standard English?

Reviewer #1: No

Reviewer #2: No

5. Review Comments to the Author

Reviewer #1: 1. the study type has been mentioned as community based descriptive study whereas bivariate and multivariate analysis has been performed. Thus, it would be better if you would change it.

2. age of the child was determined by only asking the mother, why not by asking for any documents or vaccination card?

3. The age and other variables are expressed as mean (was normality testing done). If not then it should be done and variables should be expressed accordingly

4. There are many grammatical and punctuation errors in the manuscript. Please do address them.

Reviewer #2: The topic is interesting and important from public health point of view. But changes are necessary to make it better.

Suggestions:

• There are errors in grammar and language in many places throughout the text. Please proof read and correct it

• Please spell out the numbers below ten (ex: 3 as three, 4 as four)

• Line 119: What do you mean by local units? Please write clearly.

• Line 125: Why 56 participants from each ward? Any specific reason? Steps of sampling are not coherent.

• Line 134-138: Was pretesting done to test validity or reliability?

• Line 142: Are all enumerators Muslim Or is one enumerator Muslim by religion? This is not clear.

• Line 146- 149: As the study is for children between 6 to 59 months, will these measures apply for all? How will you measure weight and height of children who cannot stand by themselves?

• Line 147: Please write clearly about the instruments used. Are “Seca digital scale and Stature meter” standard terms?

• Line 158-165: Is there any particular reason why percentage of stunting among the children is not taken an indicator for nutritional status?

• 183: Ethical not ethics

• 186: What are these levels?

• 188: Assent not ascent

• 194: Why is this necessary when the study is not school based?

• Table 1: What is the reason for classifying/categorizing age of children as ≤ 24 months and > 24 months; Number of Living Children as ≤ 2 children and > 2 children; Number of family members as ≤ 4 members and > 4 members?

• Table 2: All children are less than 5 years old, so why is the school going status necessary?

o Why is assessing “Food Sufficiency in a year from their own land” necessary?

• Table 3: Where is the data for children with normal nutritional status?

• Results: The language while writing the results are not clear at many places. You need to read the result section and rewrite some sentences.

• Discussion: Rather than just mentioning that your finding is similar to or different from a certain study, it would be better if you include added information like the type of study, sample size, pertaining results etc. from that study.

6. PLOS authors have the option to publish the peer review history of their article (what does this mean?). If published, this will include your full peer review and any attached files.

Reviewer #1: No

Reviewer #2: No

---

## [Author Response · Author response to Decision Letter 0]

19 Dec 2022

Cover Letter (Author's response to reviews)

Title: Nutritional Status and Its Associated Factors among under Five Years Muslim Children of Kapilvastu District, Nepal (Manuscript Number: PONE-D-22-29132R1) with minor revision

 Corresponding Author: Chet Kant Bhusal1*: bhusalck3112@gmail.com

Dear Academic Editor 

PLOS ONE,

Thank you very much for sending me your and the reviewers’ valuable comments on my manuscript [Nutritional Status and Its Associated Factors among under Five Years Muslim Children of Kapilvastu District, Nepal (Manuscript Number: PONE-D-22-29132R1) with minor revision.

I hope that the revised manuscript now addresses all the editor and reviewer comments. As, per your (Academic Editor) suggestion, funding related information is removed from the acknowledgements and other section of the manuscript. Now funding related information is only included in Funding Statement subsection as "“NHRC Provincial Research Grant 2078/79” having Ref no. 1090 and protocol registration no.644/2021 P". “The funders had no role in study design, data collection and analysis, decision to publish, or preparation of the manuscript". As per your suggestion, I had uploaded the data set necessary to replicate the study findings as Supporting Information files. I had included the caption for the Supporting Information files at the end of the manuscript.

Please find attached response to the comments as well as the revised manuscript with highlighted changes. Looking forward to hearing from you soon.

With Best Wishes,

Chet Kant Bhusal, MPH, MA, BPH 

Department of Community Medicine, Universal College of Medical Sciences and Teaching Hospital, Tribhuvan University, Bhairahawa, Rupandehi Nepal

Authors’ response to Editor

Response: Thank you very much for your kind suggestion. The revised version of manuscript now meets PLOS ONE's style requirements, including those for file naming.

Response: Thank you very much for finding this blunder mistake. Actually it was typo mistake and now corrected in the revised version of manuscript. 

"“NHRC Provincial Research Grant 2078/79” having Ref no. 1090 and protocol registration no.644/2021 P"

Response: Thank you very much for your kind suggestion. The funder had no any role. As per your valuable suggestion we had stated “The funders had no role in study design, data collection and analysis, decision to publish, or preparation of the manuscript" in the Funding Statement sub section in the manuscript and Cover Letter. 

"Authors would like to express their deepest gratitude to Nepal Health Research Council (NHRC Nepal) for providing “NHRC Provincial Research Grant 2078/79” having protocol number (64412021 P) for conducting this study.."

"“NHRC Provincial Research Grant 2078/79” having Ref no. 1090 and protocol registration no.644/2021 P"

Response: Thank you very much for your kind and valuable suggestion. As per your suggestion, I had removed funding related text from the acknowledgements and other section of the manuscript. Now funding related information is only in Funding Statement subsection. It is included in the Cover Letter as "“NHRC Provincial Research Grant 2078/79” having Ref no. 1090 and protocol registration no.644/2021 P"

Response: Thank you very much for your valuable suggestion. As per your suggestion, I had uploaded the data set necessary to replicate the study findings as Supporting Information files. 

Response: Thank you very much for your suggestion. As per your suggestion, I had uploaded the data set necessary to replicate the study findings as Supporting Information files. I mentioned this information in the cover letter. Hence, I humbly request you to update the Data Availability statement.

Response: Thank you very much for your suggestion. As per your suggestion, I had included the caption for the Supporting Information files at the end of your manuscript

Response: Thank you very much for your kind suggestion. We had cited and referenced the source using Endnote software. The references are complete and correct as per PLOS ONE referencing style. No any retracted citation had been done in the study. We had replaced small bracket (), by large bracket [] as per PLOS ONE style. 

Authors’ response to reviewers 

Reviewer 1: 

1. the study type has been mentioned as community based descriptive study whereas bivariate and multivariate analysis has been performed. Thus, it would be better if you would change it.

Response: Thank you very much for your valuable comment. Community based descriptive study has been mentioned as there are no any comparison group, we had conducted the study among Muslim under five children (no any other communities for comparison with Muslim) and present the finding. Bivariate and multivariate analysis has been performed in order to find the association along with OR and CI (associated factors) of nutritional status with different independent variables. We had now omitted the word “descriptive” and write “Community based cross-sectional study” in the revised manuscript. Please go through Study design and source of population sub sections under Materials and methods.

2. age of the child was determined by only asking the mother, why not by asking for any documents or vaccination card?

Response: Thank you very much for finding this mistake. Exactly we had asked the question regarding the age of mothers, family type and others during the interview, however in order to find the Nutritional status (Dependent variables) date of birth and date of interview is needed, hence the interviewer had checked the birth certificate/vaccination card after doing anthropometric measurement and record the information properly. We had used date of birth which was recorded from birth certificate or vaccination card. We had corrected it in the revised version of manuscript. Please go through second last sentence of 2.4 Anthropometric measurement. 

3. The age and other variables are expressed as mean (was normality testing done). If not then it should be done and variables should be expressed accordingly

Response: Thank you very much for your proper feedback, after your comment we had tested normality and the variables are now expressed as median and interquartile range instead of mean and SD as the data are not normally distributed shown by Kolmogorov-Smirnov test. Please go through line number of Results section of page number 9 and table number 1 of revised manuscript. 

4. There are many grammatical and punctuation errors in the manuscript. Please do address them.

Response: Huge acknowledge to the reviewer for his/her support in refining the manuscript. As per your suggestion, I had corrected whole manuscript through Native English Speaker. You can see the changes throughout manuscript. 

Reviewer #2: 

The topic is interesting and important from public health point of view. But changes are necessary to make it better.

Response: Thank you very much for your great effort, constructive feedback and time you gave me to refine this manuscript. I had addressed all of your comments according to reviewer report. As per your suggestion I had revised the manuscript with great effort. Hope it will enrich and strengthen the manuscript for publication. 

Suggestions:

1. There are errors in grammar and language in many places throughout the text. Please proof read and correct it

Response: Huge acknowledge to the reviewer for his/her support in refining the manuscript. As per reviewer suggestion, I had corrected whole manuscript through Native English Speaker. You can see the changes throughout manuscript in the revised version. 

2. Please spell out the numbers below ten (ex: 3 as three, 4 as four)

Response: By acknowledging the reviewer comment I had refined and organized the whole manuscript as per suggestion. You can see the changes throughout manuscript in the revised version 

3. Line 119: What do you mean by local units? Please write clearly.

Response: Thank you very much for this feedback. Local units regarding administrative structure of Nepal refers to (a) Metropolitian city (b) Sub Metropolitian city (c) Municipality and (d) Rural Municipality. However there are only Municipality and Rural Municipality in the study district. In the previous manuscript it was in study setting 2.6 after data processing and analysis, hence now we had moved study setting in 2.2 and rearrange all the subsection. As per your suggestion, we had corrected the sentence as “There are altogether ten local units (municipalities and rural municipalities) in Kapilvastu district of Nepal, out of which six are municipalities and four are rural municipalities”. Please go through line number 133 to 134 in the 2.3. Sample Size Determination and Sampling Technique subsection.

4. Line 125: Why 56 participants from each ward? Any specific reason? Steps of sampling are not coherent.

Response: Huge acknowledge to the reviewer for his/her support in refining the manuscript. The sample size of the study is 336. As per the sampling technique of the study, six wards were selected. Hence in order to select equal respondents from each wards we had taken equal sample size of 56. We had corrected the language and make it clearer. Please go through 140 to 141.

5. Line 134-138: Was pretesting done to test validity or reliability?

Response: Huge acknowledge to the reviewer for finding this mistake. It was corrected as reliability and we had also change the language. Please go through line number 151 to 154.

6. Line 142: Are all enumerators Muslim Or is one enumerator Muslim by religion? This is not clear.

Response: Thank you for this feedback. Now we make it clear “In each municipality, one Muslim enumerator was sent for data collection and translation”. Please go through line number 157 to 159.

7. Line 146- 149: As the study is for children between 6 to 59 months, will these measures apply for all? How will you measure weight and height of children who cannot stand by themselves? Line 147: Please write clearly about the instruments used. Are “Seca digital scale and Stature meter” standard terms?

Response: Huge acknowledge to the reviewer for finding this mistake. Previously, we had missed to incorporate the information for the children who cannot stand by themselves. Now it is added. Please go through line number 161 to 169.

Now it is corrected as “Seca digital scale was used for measuring weight and Stature meter was used for measuring height of children who were older than 24 months and can stand by themselves developmentally. Similarly, a hanging salter scale was used for measuring the weight and an inelastic measuring tape was used for measuring the height/length of the children who were less than equal to 24 months and were unable to stand by themselves. Less than equal to 24 months old children were measured while lying down. Weighing scale was standardized daily with standard weights. The weight was taken on barefoot and minimal clothes to the nearest 0.1 kg while height was taken without shoes and cap to the nearest 0.1 cm”.

8. Line 158-165: Is there any particular reason why percentage of stunting among the children is not taken an indicator for nutritional status?

Response: Thank you very much for this valuable comment. Actually this is large project and from this, we were planning to prepare another manuscript regarding the indicator stunting. However after your feedback we had incorporate the information in this manuscript to make it more strong paper. As per our knowledge there are no studies regarding nutritional status among Muslim Children. Hence we want to knock policy makers to improve nutritional status for both acute and chronic malnutrition among Muslim Children.

9. 183: Ethical not ethics

Response: Thank you for finding this mistake. It is corrected in revised version

10. 186: What are these levels?

Response: Thank you very much for this comment. Now it is corrected by specifying it as “Concerned stakeholders of municipalities and ward offices were officially contacted with letters and permission was obtained”

11. 188: Assent not ascent

Response: Thank you very much for finding this mistake. It is corrected in revised version

12. 194: Why is this necessary when the study is not school based?

Response: Thank you very much for your constructive feedback. After your feedback we removed this irrelevant information. Please check it.

13. Table 1: What is the reason for classifying/categorizing age of children as ≤ 24 months and > 24 months; Number of Living Children as ≤ 2 children and > 2 children; Number of family members as ≤ 4 members and > 4 members?

Response: 

(a) Reason for categorizing age of children as ≤ 24 months and > 24 months. 

From birth to 24 months refers to “The Golden 1000 Days” from conception (230 days) to 2 years (265+ 265=730 days). The "Golden 1000 Days" initiative, sponsored by UNICEF and run by the Ministry of Health and Population of Nepal, indicated that there is a serious, lifelong, and irreparable nutritional crisis among young children under the age of two years (NationalPlanningCommision, 2016; SaveTheChildren, 2012). Similarly, 80% of the human brain development take place during this "Golden 1000 Days” (NationalPlanningCommision, 2016). Hence this study was intended to calculate the association between nutritional status among children of ≤ 24 months and > 24 months.

(b) Reason for categorizing Number of Living Children as ≤ 2 children and > 2 children; 

According to Nepal Demographic and Health Survey 2016, Total Fertility rate in Nepal was 2.3 and in NDHS 2022 it is 2.1, hence taking this TFR as reference, this study categorized Number of Living Children as ≤ 2 children and > 2 children in the study.

(c) Reason for categorizing Number of family members as ≤ 4 members and > 4 members?

According to the Central Bureau of Statistics (CBS) 2011, average family size is 4.88 members and as per Annual Household Survey 2015/16 the average size of the family in Nepal is 4.6 person at the national level. Hence taking this average family size as reference, this study categorized family members as ≤ 4 members and > 4 members.

14. Table 2: All children are less than 5 years old, so why is the school going status necessary?

o Why is assessing “Food Sufficiency in a year from their own land” necessary?

Generally less than 5 years children refers to 6 months to 59 months, which also includes the school going age of the children as several studies conducted in Nepal (Tharu, 2017), https://edusanjal.com/blog/education-levels-nepal/ found that majority of the children start their schooling around the age of 3 years in Nepal. Many schools are conducting programs such as (a) periodic medical checkup by pediatrician doctors, (b) distribution of meal for 2-3 times in many schools as per dietician instruction which ultimately affects in the nutritional status. Similarly, Government of Nepal is providing Meal Day school program for government schools. Hence, as the study is community based so authors assumed that few Muslim parents might send their children to such schools, thus school going status is taken in the study. 

Nepal is a agricultural country (MoAC, 2005; Paudel, 2016) and many people depends on agriculture for their food. About 20% of food self-insufficient households were unable to fulfill the minimal income need for food security during the month of shortfall (Joshi & Maharjan, 2007). On the top of that The Muslim populations in Nepal was given a lower marginal status in the legal code, and they were shunned, discriminated against, and attacked violently (Sijapati, 2011)

15. Table 3: Where is the data for children with normal nutritional status?

Response: Thank you for this feedback. Actually, WHO Anthro Software does not measure overall normal nutritional status like that in the MUAC (Mid Upper Arm Circumference). MUAC only gives SAM and MAM. Hence, in order to find the nutritional status (Underweight, overweight, wasting and stunting) researcher used WHO Anthro Software which is more valid. However after reviewer valuable comment, we had presented the information as Underweight –Yes and No, Overweight-Yes and No, Wasting-Yes and No, similarly Stunting- Yes and No in the table 3. Please go through line number 239-240.

16. Results: The language while writing the results are not clear at many places. You need to read the result section and rewrite some sentences.

Response: Huge acknowledge to the reviewer for his/her support in refining the manuscript. As per your suggestion, I had corrected it through Native English Speaker. You can see the changes throughout the manuscript in the revised version. 

17. Discussion: Rather than just mentioning that your finding is similar to or different from a certain study, it would be better if you include added information like the type of study, sample size, pertaining results etc. from that study.

Response: Thank you very much for this comment. I had added the information as per your suggestion with great effort. Please go throughout the discussion portion. You can see the changes throughout the discussion in the revised version. 

Thank you very much for all the editor and reviewer for their great effort, constructive feedback and time they gave me to refine this manuscript. I had addressed all of their comments according to reviewer report. As per their suggestion I had revised the manuscript with great effort. Hope it will enrich and strengthen the manuscript for publication. I would like to acknowledge with sincere gratitude to all the editors and reviewers for their efforts to improve my manuscript. Looking forward to hearing from you soon. Thanking You.

---

## [Editor Report · Decision Letter 1]

27 Dec 2022

Nutritional Status and Its Associated Factors Among Under Five Years Muslim Children of Kapilvastu District, Nepal

PONE-D-22-29132R1

Dear Dr. BHUSAL,

We’re pleased to inform you that your manuscript has been judged scientifically suitable for publication and will be formally accepted for publication once it meets all outstanding technical requirements.

Kind regards,

Reshu Agrawal Sagtani

Academic Editor

PLOS ONE

---

## [Editor Report · Acceptance letter]

5 Jan 2023

PONE-D-22-29132R1 

Nutritional Status and Its Associated Factors Among Under Five Years Muslim Children of Kapilvastu District, Nepal 

Dear Dr. Bhusal:

I'm pleased to inform you that your manuscript has been deemed suitable for publication in PLOS ONE. Congratulations! Your manuscript is now with our production department. 

Kind regards, 

on behalf of

Dr. Reshu Agrawal Sagtani 

Academic Editor

PLOS ONE